# Recent Advances and Challenges in Schumann Resonance Observations and Research

Jinlai Liu [1,2], Jianping Huang [1,2,3,*], Zhong Li [3], Zhengyu Zhao [4], Zhima Zeren [2], Xuhui Shen [5] and Qiao Wang [2]

1   School of Emergency Management Science and Engineering, University of Chinese Academy of Sciences, Beijing 100049, China; liujinlai21@mails.ucas.ac.cn
2   National Institute of Natural Hazards, Ministry of Emergency Management of China, Beijing 100085, China; zerenzhima@ninhm.ac.cn (Z.Z.); qiaowang@ninhm.ac.cn (Q.W.)
3   School of Information Engineering, Institute of Disaster Prevention, Langfang 065201, China; lizhong@cidp.edu.cn
4   School of Electronic Information, Wuhan University, Wuhan 430072, China; zhengyuzhao@whu.edu.cn
5   State Key Laboratory of Space Weather, National Space Science Center, Chinese Academy of Sciences, Beijing 100190, China; shenxuhui@nssc.ac.cn
*   Correspondence: jianpinghuang@ninhm.ac.cn

**Abstract:** The theoretical development of Schumann Resonances has spanned more than a century as a form of global natural electromagnetic resonances. In recent years, with the development of electromagnetic detection technology and the improvement in digital processing capabilities, the connection between Schumann Resonances and natural phenomena, such as lightning, earthquakes, and Earth's climate, has been experimentally and theoretically demonstrated. This article is a review of the relevant literature on Schumann Resonance observation experiments, theoretical research over the years, and a prospect based on space-based observations. We start with the theoretical background and the main content on Schumann Resonances. Then, observations and the identification of Schumann Resonance signals based on ground and satellite data are introduced. The research and related applications of Schumann Resonances signals are summarized in terms of lightning, earthquakes, and atmosphere. Finally, the paper presents a brief study of Schumann Resonances based on the China Seismo-Electromagnetic Satellite (CSES) and preliminary ideas about how to improve the identification and application of space-based Schumann Resonances signals.

**Keywords:** Schumann resonance; natural phenomena; lightning; CSES; background characteristics

## 1. Introduction

The Earth–ionosphere cavity is composed of the Earth's surface with high conductivity, the conductive but dissipative ionosphere, and an insulating air layer in the middle. The cavity formed between the Earth's surface and the ionosphere allows for the presence of quasi-electromagnetic standing waves, whose wavelengths are comparable to that of interplanetary waves [1]. In the lower atmosphere, up to 60–70 km above the Earth's surface, the limited electrical conductivity of the atmosphere is mainly maintained by cosmic rays [2]. Near the ground, the conductivity is around the order of 10–14 S/m, thus making this area a good electrical insulator. The conductivity increases exponentially with height on a characteristic scale of 3–6 km, from the ground to 75–95 km below the E layer, and its specific value depends on local time and height. Above this height, the ratio of the electron cyclotron frequency to the electron-neutral particle's collision frequency cannot be ignored, and the conductivity becomes a tensor [3,4]. In the upper part of the D layer and the lower part of the E layer, the conductivity component parallel to the magnetic field is about $10^{-4}$~$10^{-2}$ S/m, which can be compared with the conductivity magnitude of the Earth's surface and water. For ELF waves (3–300 Hz), the transmission between the insulated lower atmosphere and the ionosphere occurs at 40–50 km [5]. For the lower

atmosphere, the main effect is electron displacement, while for the ionosphere, conduction is the major influence [6].

Some strong electrical transient processes, such as lightning and electromagnetic radiation pulses, extend from the excitation source into the cavity. This process was studied by many scientists through experimental simulations of conducting concentric spheres in the late 18th and early 19th centuries. In 1893, George Francis Fitz Gerald [7] first proposed the hypothesis of Earth's natural electromagnetic resonance based on the early concentric conductive sphere experiment and the atmospheric conductive layer hypothesis at that time. In 1894, Joseph Larmor [8] calculated and derived the formula for the free period in a uniform spherical capacitor, which was identical to the formula derived by Schumann in 1952 but was not applied to the Earth–ionosphere cavity at that time. In 1921, Charles T.R. Wilson [9] proposed the concept of a global atmospheric circuit, where positive charges are transmitted through conduction to the upper atmosphere of the Earth's surface and ionosphere to form a discharge current, while negative charges are transmitted to the Earth's surface through cloud to ground lightning strikes to generate a charging current. The concept of global atmospheric circuits has rapidly promoted research on Earth–ionosphere waveguides and SR-lightning field sources. For example, Schelkunoff, Rydbeck, and others [10,11] studied the wave propagation problem of concentric conductive spheres and deduced the wave function and resonance frequency equation between concentric conductive spheres, but they did not clearly calculate the solution of the equation, nor connect the conclusion with Earth–ionosphere waveguides. The above research approached the core of the Schumann Resonance (SR) phenomenon, providing experimental and theoretical support for the later establishment of SR theory.

In 1952, Winfried Otto Schumann [12,13] deduced the formula of the characteristic frequency of the Earth–ionosphere cavity and pointed out that SR propagates in the narrow dielectric interface between the surface and the ionosphere, and the height of the interface is far less than the Earth's radius. From 1952 to 1957, Schumann [14–19] proposed the resonance theory of ELF waves excited by lightning discharge in the Earth–air–ionosphere system waveguide. His theory consists of three main parts: (1) electromagnetic wave propagation in a spherical cavity, (2) the Earth–air–ionosphere system as a waveguide, and (3) lightning discharge as an excitation source. The ELF (extremely low-frequency) resonance theory was first published by Schumann and named by Charles Polk after him [20], which is now called SR.

SR is a natural electromagnetic resonance phenomenon, which is mainly excited naturally by global lightning discharges and propagates by oscillations back and forth in the Earth–ionosphere cavity. The lowest frequency component of electromagnetic impulse (thunderstorms and high-altitude explosions, etc.) can orbit the Earth several times before undergoing decay. During the process, phase increase and wave cancellation generate a resonance spectrum, where wave cancellations always follow multiple paths [21,22]. The main characteristics of this resonance spectrum can be explained by the quasi TEM wave theory, where all resonance spectra are not correlated with the superposition of the global thunderstorm effect [23]. SR can be observed in many different places and can be detected anywhere in interstellar space. The strength of the electrical and magnetic components of SR is very weak, and it is easily mixed with nearby lightning and a large amount of unrelated human electromagnetic noise. However, in addition to lightning and artificial electromagnetic noise sources, SR is an important component of the natural background electromagnetic noise spectrum from 5 to 50 Hz [1]. For the SR effect, a more general approach is to study the propagation of ELF waves.

Madden and Thompson [24] made a great contribution to ELF wave propagation theoretical models in the 1960s. First, they pointed out that the main dissipation of SR transmission occurs in the D region of the low ionosphere, and considered the problem of leakage in the Earth–ionosphere cavity (limited to nighttime) for the first time. In studying the impact of nuclear explosions on ionospheric disturbances, an equivalent circuit model was designed to attempt to solve the SR problem. The dissipation problem of

SR/ELF waves was later independently studied by Greifinger et al. [25]. Greifinger et al. derived an approximate expression for the TEM eigenvalues of ELF waves propagating in the Earth's ionospheric waveguide, ignoring the anisotropy caused by the Earth's magnetic field. Greifinger also indicated that the eigenvalues are mainly determined by the properties of the ionosphere near the two clear heights where the maximum Ohmic dissipation occurs. Mushtuk and Williams [26] proposed a knee model for simulating a uniform earth–ionosphere waveguide based on Greifinger et al. that could accurately simulate experimental observations within the SR frequency range; the model showed that as the SR frequency increases, the Q factor also increases. The knee model mainly considers the transition from ion-dominated to electron-dominated conductivity in the lower characteristic layer of the ionosphere. Williams, Mushtak, and Nickolaenko et al. [27] compared five uniform cavity models with significant differences in waveguide dissipation behavior, and pointed out that the Lorentz process is suitable for estimating the adequacy level of these propagation parameter models, and that the knee model variables have a statistically significant and physically consistent contrast between the estimated solar activity minima and solar activity maxima for the upper magnetic layer.

The transmission and dissipation processes of SR are different during the day and nighttime. Due to the need for studying the asymmetry of day and night, Pechony and Zhou H et al. [28,29] used a non-uniform empirical "knee" model for distinguishing day and night based on the classic "knee" model. Madden and Thompson [24] provided information on the energy dissipation of EM waves in the first SR frequency band of the ionosphere with altitude variations in mid-latitude daytime and night states as early as 1965. In a cavity with day–night asymmetry, assuming that the termination line passes through the two poles and splits the Earth into two equal hemispheres, the conductivity suddenly changes at the boundary between the day–night hemispheres without any tilt, which is sufficient to evaluate the relative impact of day–night asymmetry and source migration on SR parameters.

The determination of basic theories has rapidly promoted the progress of SR research. It is now widely believed that intra-cloud lightning and cloud-to-ground lightning release is the primary source of SR, with a global average thunderstorm rate of 100/S. The peak energy released by it reaches 20,000–30,000A, with an average distance of 3–5 km, which is sufficient to observe SR for exciting the Earth–ionosphere cavity. Lightning discharges, the primary natural source of SR, occur primarily in three areas of global thunderstorm activity, including South America, Africa, and Southeast Asia [30,31]. In addition, local dense lightning can also excite SR. Simultaneous observation of SR anomalies in France, Italy, Russia, and Japan were reported during the 2022 Tonga eruption, where localized lightning field sources were formed by aggregation in the eruption cloud [32]. J. Bor et al. [33] used the data observed by multiple SR and PG (potential gradient) measurement stations distributed globally to study the GEC's response to the eruption of the Tonga volcano. SR and PG intensities were used to describe the AC (alternating current) and DC (direct current) components of the GEC, respectively. The SR measurements indicated that the impact of lightning activity in the eruption is global. The DC component of the GEC was charged twice during the eruption by cloud-to-ground lightning strikes. Nickolaenko et al. [34] used the vertical electric field and two orthogonal horizontal magnetic field components recorded from multi-station SR observations to calculate the Umov–Poynting vector (UPV), which can be used to locate and estimate eruptive activity. During the eruption of the Tonga volcano, charged dust and dust plasma may have discharged during charge accumulation, forming local lightning sources under certain circumstances, which in turn excite SR, but have no significant effect on real global thunderstorm activity [35,36]. Analyzing the geomagnetic field disturbance and SR response during the 2022 Tonga eruption, Gavrilov et al. [37] concluded that Tonga volcanic activity generates global geomagnetic disturbances via acoustic–gravity waves (AGW), which leads to changes in ionospheric conductivity, ionospheric current values, and the geomagnetic field; the analysis also revealed that SR signal properties are related to the number and energy of lightning discharges.

Qiaoli Kong et al. [38] studied vertical ionospheric disturbance types during volcanic activity and classified the origins of the disturbances as acoustic–gravity wave, Lamb wave, and tsunami wave.

When lightning activity occurs, the lightning channel is similar to a huge electromagnetic wave transmitting antenna, emitting electromagnetic energy with a radiation frequency below 100 kHz. Although lightning signals below 100 Hz are very weak, their attenuation is only 0.5 dB/mm, so the electromagnetic waves generated by a single lightning activity discharge can propagate multiple times globally and then attenuate into background noise [39]. When the wavelength of the ELF wave is equivalent to the circumference of the Earth (about 4000 km), electromagnetic waves propagating in the opposite direction in the Earth–ionosphere cavity interfere with the straight beam excited by lightning at the SR frequency, amplifying the spectral signal from lightning and ultimately leading to the appearance of harmonic peaks of various modes of SR in the ELF spectrum.

SR signals propagate within the Earth–ionosphere cavity, where the cavity between the Earth and the ionosphere can be seen as a waveguide with infinite horizontal width but limited vertical depth. For cavities with ideal conductive boundaries, the SR eigenfrequency $f_n$ is approximately given by the average radius a of the resonant cavity and the speed of light (Equation (1)).

$$f_n = \left(\frac{c}{2\pi a}\right)\sqrt{n(n+1)} \tag{1}$$

By substituting the specific value of n, the SR frequency without loss can be obtained: $f_1 = 10.6$ Hz, $f_2 = 18.3$ Hz, $f_3 = 26.0$ Hz, etc. The SR eigenmodes based on digital simulation also showed the same results [40].

Due to the resonance effect, SR mainly propagates in a transverse magnetic field (TM) mode within the cavity, with significantly higher transmission energy than other frequencies [41]. The loss of electromagnetic waves in the real Earth–ionospheric cavity medium is mainly propagated in the D region of the low ionosphere at the system boundary. The loss in the medium of the real Earth–ionosphere cavity leads to a broadening of the resonance lines and lowering of the resonance peak frequencies in comparison with the "ideal case" [42].

## 2. Observation Methods

### 2.1. Ground Observation

Ground-based observation has always been a common method for obtaining SR electromagnetic data and conducting related research. In 1960, Martin Balser and Charles A. Wagner [43] first published ground-based observations characterizing the first five eigenmodes of SR, as shown in Figure 1.

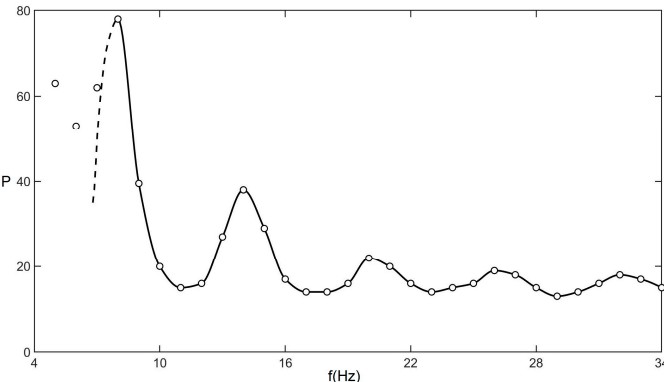

**Figure 1.** The first publicly composite published spectrum of Schumann Resonances. The figure is readapted with permission from Ref. [43]. 1960, M. Balser et al.

With the popularization of radio technology and the rapid development of software and hardware, relevant research teams have been designing and manufacturing receiving

systems for SR-related research. For example, Ohta et al. designed a ULF/ELF observation system that utilizes narrowband sensors to receive signals within the SR frequency band [44]. The sensor's signal records can display characteristics that are not present in amplitude changes generated by broadband sensors, which is conducive to real-time monitoring of SR anomalies before earthquakes.

In recent years, with the development of technology, devices used to collect SR signals have gradually become portable and diverse. Votis et al. [45] independently designed a portable SR receiver and received stable SR signals over a continuous period of time. Cano Domingo [46] added a sensor with the average center frequency of the first eigenmode of SR as the focus in the portable acquisition device to broaden the system response to extract transient events with a higher resolution. Tatsis et al. [47] designed a prototype testing fixture for calibrating SR measurement equipment to test the accuracy and stability of SR antenna receivers. In addition, the background of SR is also shown in the gravitational wave measurement results of the interferometer [48,49]. During the period of China's tenth five-year plan, China established the "China Digital Seismic Observation Network" project, in which 12 very low-frequency observatories continuously observe the natural electromagnetic field from 0.1 to 800 Hz according to a unified specification, providing important data support for ELF signal propagation and seismic-prediction-related research [50].

ELF electromagnetic data processing and SR extraction and analysis methods have also developed rapidly. Based on the nonlinear interaction model between SR signals and HF electromagnetic waves under the action of atmospheric electric fields, Cao B.X. et al. [51] used the first batch of SR observation stations in China to obtain the SR spectrum through short-wave timing signal BPM demodulation. Rodríguez Camacho et al. [52] determined the amplitude spectrum from the measured low-amplitude noise signal and manually denoised it. Finally, they calculated the analysis function to fit the filtered amplitude spectrum and extracted accurate SR parameters. Soler Ortiz [53] proposed a new method for statistical analysis of SR signals in the time domain. By using the maximum likelihood parameter estimation and distribution fitting of the Akaike Information Criterion (AIC), statistical analysis is applied to characterize the time series of SR and the bandwidth of their ELF spectrum. Salinas et al. [54] combined the Lorentz function and linear term to perform nonlinear fitting on the measured ELF spectrum to identify the existence of SR and quantitatively characterize it. They used the Levenberg–Marquad algorithm to optimally constrain the measured SRs and extract the SR parameters with high confidence.

## 2.2. Satellite Observation

Compared with ground observation, increasing the observation height is beneficial for receiving more abundant electromagnetic information. In 1977, T. Ogawa [55] observed the first seven eigenmodes of SR signals at stratospheric altitude using high-altitude balloons. With the development of space technology, satellite payloads are more sensitive and precise, and electromagnetic satellite monitoring has gradually become the main means of space-physics-related research. Zhao Z.Y. et al. [56] used the ELF electromagnetic observation data of Aureol-3 near Earth's polar orbit low-altitude satellite on ionospheric electric field disturbance and electron density disturbance at the height of ionosphere layer F, observed SR signals in the electric field component and showed that the SR eigenfrequency occurring in the upper ionosphere is related to the large-scale irregularity and positive density gradient of electron density, which is the first SR research based on satellite data. Later, Ni B.B. and Zhao Z.Y. [41] pointed out that the electric field component observed by the Aureol-3 satellite has a good resonance spectral structure, and the peak frequency corresponds to the various SR eigenfrequencies in the magnetic field component, confirming that the ELF wave field disturbance observed by the Aureol-3 satellite is an electromagnetic oscillation related to SR. In 2011, Simões et al. [57] detected and extracted SR signals at the ionosphere F-layer height using electric field data collected by the C/NOFS satellite during abnormally low solar activity minima. The C/NOFS satellite observes ionospheric electromagnetic signals at the height of the ionosphere F-layer, and can observe significant

SR phenomena under nighttime conditions. In 2014, Dudkin et al. [58] extracted characteristic records of ionospheric Alfven resonance and SR using the ULF/ELF electric field observation data of the Chibis-M satellite. Simões and Dudkin [57,58] extracted SR signals based on satellite altitude observations, which proved the spatial SR leakage mentioned by Madden and Thompson [24]. Toledo Redondo et al. [59] used more than four years of electric field data sets from the DEMETER satellite to produce detailed maps of the effective ionospheric reflection height or D-region height, which is inversely proportional to the cutoff frequency. The depiction of the Toledo-Redondo et al. map ranges within 60° north–south latitudes, while the global SR background field based on CSES is within 65° north–south latitudes. The SR background and Toledo-Redondo's maps both apparently shows anomalous enhancement due to geographic and geomagnetic misalignments. Toledo's research can support subsequent studies of SR context based on satellite data.

### 3. Schumann Resonance and Lightning

SR is closely related to global lightning and can provide useful information about global thunderstorm activity and low ionospheric parameters. SR frequency variation is one of the important indicators of global lightning distribution [60]. Lightning generates electromagnetic fields and waves in all frequency ranges, and in the ELF range below 100 Hz, frequency eigenmodes such as 8 Hz, 14 Hz, and 20 Hz can excite global SR signals [39].

Ouyang X.Y. et al. [30] used SR observation data from the Yunnan Observatory in China to analyze the background variation characteristics of SR signals; their results indicated that the resonance frequency, amplitude, and spectral integral intensity of SR signals exhibit certain daily, monthly, and seasonal variation patterns with global lightning development. Yin F., Zhang Q.L. et al. [31] analyzed the response characteristics of various SR eigenmodes to the three major global flash power sources using continuous ELF data observed at Israel's Mitzpe Ramon (MR) station and OTD/LIS satellite lightning data. Tatsis [47,61] extracted and analyzed the transient signal features of the ELF spectrum, discussed the relationship between this feature and local lightning activity, and summarized the diurnal and seasonal variations of the first five eigenmodes of SR. Koloskov et al. [62] used polar long-term SR monitoring data to study global thunderstorm activity changes and identified the main factors affecting the seasonal and annual variation characteristics of SR eigenmodes.

In recent years, SR measurements have been widely used to study global lightning localization and distribution. The lightning localization problem can be solved using two methods: multi-station localization and single-station localization [39,63]. Multi-station technology is more precise but requires more complex and expensive facilities, such as direction finders or arrival time sensor networks. In order to extract lightning intensity and position information from SR records accurately, single-station positioning technology must consider the source observer distance (SOD) [64]. A. V. Shvets et al. [65] used simultaneous SR observation data of multiple stations to obtain the intensity–distance distribution and spatial distribution of lighting sources at the stations through two-stage inversion. Yamashita et al. [66] incorporated a time-of-arrival (TOA) method for global geolocation and charge moment estimation of lightning and transient luminescence events based on a network of four ELF stations. E. Prácser et al. [67] used multi-station SR observation data to invert and reconstruct global lightning activity. Boldi et al. [68] proposed an inversion model based on multi-station SR measurement, which can calculate the average charge moment change per second from SR receiving stations as an important indicator of global lightning activity distribution and intensity. Progress has been made in lightning inversion methods based on VLF and ELF waves in the Earth–ionosphere cavity, and the inversion methods and effects in these two frequency bands are different [69].

Utilizing the connection between SR and lightning activity for large-scale lightning activity research requires the use of various lightning activity data [70]. Currently, common large-scale lightning activity data include the OTD/LIS satellite data, the FY-4A satellite LMI (Lightning Mapping Imager) data, and the WWLLN global lightning positioning

network, among others. The OTD/LIS satellite data is a climate data product that describes global lightning activity, with a time span of 11 years (May 1995 to April 2006) [71]. FY-4A, a Chinese geostationary meteorological satellite, was launched in December 2016 and carries LMI that enables continuous observation of total lightning including cloud flashes, intercloud flashes, and ground flashes over the Asia and Oceania regions [72]. WWLLN (World Wide Lightning Location Network) is composed of very low-frequency (VLF) radio lightning sensors, which calculate lightning activity location data worldwide by observing pulse signals from lightning discharges in the 3–30 kHz frequency band [73].

SR can also be used to detect and study upper atmospheric discharges (transient lightning) and extraterrestrial lightning. Research has found that strong lightning discharge can lead to transient luminescence events such as sprites and elves [74,75], resulting in ELF transients [76–79]. Q-bursts are the ELF transient events produced by the sequence of direct and antipodal pulses from a source lightning stroke occurring around the world [80]. Toshio Ogawa et al. [81,82] promoted the naming of the Q-burst phenomenon and made an important contribution to the study of the mechanism of Q-burst generation and propagation. J Bór et al. [83] analyzed the deviation between the source azimuth of Q-bursts estimated from ELF data and the true luminescence source azimuth, and obtained accurate Q-burst azimuth estimates by correcting the azimuth deviation due to local effects.

In addition, many celestial bodies in the solar system other than Earth, such as Venus, Mars, Jupiter, Saturn, and Titan, can naturally generate SR through lightning activity [84]. Scholars use observational data from interstellar or planetary orbit probes to conduct research on extraterrestrial lightning based on these celestial bodies [85,86]. However, due to a lack of understanding of the electromagnetic environment of extraterrestrial celestial bodies, there are currently limited methods for studying the field of extraterrestrial lightning based on SR theory [87,88].

Short-term events such as lightning and earthquakes can cause changes in ionospheric cavity properties, such as ELF transients, and these change characteristics can be identified in the SR band. Therefore, in future monitoring of lightning, earthquake, and solar radiation events, in addition to the traditional monitoring means, related scholars can use strong ELF transient emission from mesoscale convective systems as a kind of long-wavelength radar to repeatedly detect ionospheric properties and collect transient changes of SR signals and their patterns.

## 4. The Application of Schumann Resonance

On the basis of natural lightning field source excitation, SR, a natural signal, is mainly affected by energy lines, acoustic noise, rain, dust, trains, cars, and other disturbances. The frequency, amplitude, and half-width of SR exhibit abnormal changes at each eigenmode [39]. Based on the research progress of SR measurement equipment and analysis methods, the propagation process and influence mechanism of SR have become a new research hotspot. The SR signal is mainly influenced by solar activity and global lightning activity, ultimately exhibiting corresponding daily, seasonal, and annual variations [41,89–91].

Solar activity indirectly affects the SR signal by perturbing the Earth–ionosphere cavity. A two-dimensional telegraphic equation (TDTE) transmission line model of the Earth–ionosphere waveguide was developed by A. Kulak et al. [92], which can be used to calculate the decay rate of the SR frequencies relative to the theoretical values. The decay rate of the Earth–ionosphere cavity was calculated by A. Kulak et al. [93] using daily observations of the N-S and ELF magnetic field for the period from the minimum to the maximum of solar cycle #23. It is concluded that the first SR frequency increases with the increase in the solar activity level and the daily average decay rate decreases. In addition, the Earth–ionosphere cavity is affected by the long-term variation of the solar activity with a corresponding change, which has a delay of about 1 year with respect to the variation in the solar activity level. Sátori et al. [2,94] suggested that the solar X-ray flux dominates the E region and has a negligible effect on the D region, which can be explained by an electric and magnetic heights model for the lower ionosphere.

In the response to solar events, the SR response is more pronounced in frequency than in amplitude, while the SR frequency increases non-monotonically with the X-ray flux due to the overlapping effect of protons. There are long-term deformations in the Earth–ionosphere cavity. Bozóki et al. [95], based on a study by Toledo-Redondo et al. [59] on the effective reflection height of the Earth–ionosphere cavity calculated using DEMETER measurements combined with multisite SR measurement records, identified the sources responsible for these deformations (X-ray-dominated deformation of the cavity at low latitudes and energetic particle precipitations (EEP) dominate the deformation of the cavity at high latitudes). Kudintseva et al. [96] pointed out that during sudden ionospheric disturbances (SIDs) caused by X-rays and solar flares, SR anomalies can indicate changes in the conductivity of the middle atmosphere. Recent studies have shown that SR may overlap with ionospheric Alfven resonance in the same frequency band (1–30 Hz), and may also be affected by geomagnetic disturbances [97,98].

SR anomalies can provide rich electromagnetic information. Relevant scholars are committed to analyzing and summarizing the background characteristics of SR, extracting and analyzing abnormal disturbances caused by seismicity, natural climate change, etc., and hoping to establish the relationship between SR anomaly and natural phenomena of Earth.

### 4.1. The Correlation of Anomalies in Schumann Resonances and Earthquakes

Earthquakes and pre-earthquake activities are often accompanied by electromagnetic activity anomalies, which may be caused by seismogenic effects such as the release of ionized gases before earthquakes, electric field disturbances caused by atmospheric gravity waves (AGW), and infrasound turbulence excited after earthquakes [99,100]. Earthquake electromagnetic radiation can pass through rocks and soil with low attenuation in the ELF frequency band, enter the air from the source, and cause SR anomalies [101].

Ohta et al. [102] used ELF observation data from the Nakatsugawa observation station in Japan from early 1999 to the end of 2004 to analyze the relationship between SR anomalies and multiple earthquakes in Taiwan during the observation period. The results showed that almost all land earthquakes can observe corresponding SR anomalies, and the anomalies always occur about a week before the earthquake. De. S. S. et al. [103] studied the changes in SR signals 15 and 5 days before and after earthquake events, and concluded that SR anomalies appeared several hours earlier than ionospheric anomalies. A. P. Nickolaenko, M. Hayakawa et al. [104–106] used ELF observation data from Nakatsugawa and Moshiri stations in Japan to report significant SR anomalies before and after multiple earthquakes and found that the estimated arrival angle of the abnormal signal was consistent with the azimuth angle from the station to the epicenter, proving that seismic electromagnetic radiation is the main cause of SR anomalies. Christofilakis et al. [107] used the close ELF observation records obtained during the earthquake duration (the ground observation station is only 365 km from the epicenter) to detect the extreme electromagnetic events before and after the occurrence of seismicity and summarized the laws of these extreme electromagnetic events. Galuk et al. [108] established a numerical model of ELF radio wave scattering using the theory of local non-uniformity seismogenesis theory, which can serve as a reference for estimating and correcting earthquake magnitudes. Sierra Figueredo et al. [109] used ELF observation data from Mexico to extract and statistically analyze anomalies in the first three eigenmodes of SR for 12 earthquakes within three time windows. The statistical results showed the time distribution of SR frequency and amplitude anomalies. Tritakis et al. [110] emphasized the effectiveness and premise of ELF electromagnetic records in the SR band as earthquake precursors. Specific SR modulation is very useful for predicting seismicity, but the final prediction decision can only be completed with the help of additional observation of adjacent effects.

There are various explanations provided by relevant researchers for the causes of earthquake SR anomalies. M. Hayakawa et al. [105,106] proposed that the abnormal disturbance of SR's high-order eigenmode was caused by the difference in path length between direct

electromagnetic waves from the global thunderstorm center and non-uniform scattering electromagnetic waves from the epicenter. Recently, they used seismogenic disturbances with lower ionospheric conductivity to explain the SR anomaly, indicating that the SR anomaly is attributed to the compression and expansion of the vertical ionospheric conductivity profile above the epicenter of the earthquake [111]. Zhou H. et al. [112] used ELF observation data from two seismic stations in China to simulate the asymmetric surface ionospheric cavity between day and night using a partially uniform "knee" model of vertical conductivity profiles and introduced a local earthquake-induced atmospheric conductivity disturbance model. The simulation results showed that the SR anomaly before the earthquake may be caused by the ionospheric irregular body located above the epicenter in the crust of the ionospheric cavity.

At present, the mechanism of earthquake–SR anomalies is not clear enough, and the methods of earthquake electromagnetic monitoring are not mature enough. It is difficult to distinguish whether abnormal signals are earthquake precursors or disturbances using existing theories. However, many research results have shown that the characteristics of SR anomalies obtained from analysis and statistics before and after earthquakes tend to be consistent. The SR amplitudes measured at the same stations are enhanced before and after the earthquake under the action of an approximate excitation source. The enhancement is positively correlated with the earthquake magnitude and negatively correlated with the source–observer distance. The SR anomaly characteristics are related to the magnitude of the earthquake. When the magnitude of the earthquake is large enough (magnitude > 6.0), the SR anomaly appears within a few days to a week before and after the earthquake, and the SR anomaly appears several hours earlier than other ionospheric anomalies before the earthquake [104,111,112]. Earthquake–SR anomaly research has great potential in the field of earthquake electromagnetic monitoring and can provide new ideas and methods for earthquake prediction.

### 4.2. Application of Schumann Resonance in Other Fields

SR can provide global climate information continuously, reliably, and inexpensively, and can monitor processes that affect and are affected by the global climate over long periods of time. SR plays an important role in global climate research. Since the 1990s, the connection between lightning activity and Earth's climate has sparked research on SR-related climate research. Williams [113] pointed out that SR can monitor global temperature because the connection between SR and temperature is the lightning flash rate, which increases nonlinearly with temperature. The nonlinear relationship between lightning and temperature can serve as a natural amplifier for temperature changes, making SR a sensitive "thermometer". The daily and seasonal variations in the SR dataset also show a strong positive correlation between surface temperature and SR power [114]. In addition to indicating temperature changes, SR has also been proven to be related to the El Niño and Southern Oscillation (ENSO) phenomenon, which has a significant impact on the Earth's climate [115,116]. During the ENSO period, the warming of seawater in the central and eastern equatorial Pacific Ocean and the slight cooling of seawater in the west directly affects tropical atmospheric circulation. In the western Pacific Ocean, it is mainly the effect of the location and intensity of the subtropical high-pressure activity. ENSO affects atmospheric circulation and thus leads to significant climate anomalies on a global scale, such as persistent heavy rainfall, flooding, widespread drought, and strong storm activity [117]. Williams and Bozóki [118,119] analyzed two major El Niño events in 1997–1998 and 2015–2016. The results show that SR intensity and lightning activity were significantly enhanced during the transition from the cold to warm phase. The SR intensity may be a precursor to the occurrence of the global surface temperature maxima. The Madden Julian Oscillation (MJO), a quasi-periodic atmospheric model, primarily alters rainfall patterns in the equatorial region from Africa to the Pacific. Beggan, C.D. and Kozlova, and A.V. [120,121] attempted to correlate using SR records and MJO, but the correlation has not been explicitly confirmed.

Intense explosions occurring in the atmosphere, such as the high-altitude nuclear explosions on July 9 of 1962, caused SR transient variations including decreased SR frequencies by disturbing the Earth–ionosphere cavity [24,122]. The effect decayed away in a period of hours with γ irradiation and neutron decay of the nuclear explosion. Therefore, before the international release and implementation of *the Partial Nuclear Test Ban Treaty* in 1963, it was believed that SR could be used to monitor enemy nuclear explosion tests in remote areas of the Earth. ELF waves have low attenuation properties in the SR band and were widely used in the last century for long-distance information transmission, such as long-distance submarine communications where such waves were artificially generated [123,124]. The ELF band has a very low frequency, and the antenna emitting this wave must be very long (>200 km) for efficient transmission, so these transmitters require very high output power, and the transmission efficiency will also be very low. However, due to the ability of the signal to spread globally, superpowers still use these large ELF transmitters.

SR signals may also affect the physiology, psychology, and behavior of organisms, and organisms are particularly sensitive to changes in low-frequency electromagnetic fields, especially SR signals [125,126]. The study of SR and biological mechanisms is considered controversial, and it is necessary to identify the mechanisms and pathways of ELF electromagnetic fields affecting biological health, which has not been fully studied so far [127]. Recent studies have shown that weak magnetic fields in the SR band have a certain protective effect on mouse hearts under pressure conditions [128]. Based on statistical methods of epidemiological analysis and contemporaneous SR recordings, Fdez-Arroyabe [129] found that the power amplitude of SR has an effect on cardiovascular-related diseases in humans. In addition, Fdez-Arroyabe et al. [130] provided a glossary of terms related to atmospheric electromagnetic biological effects, which is important to facilitate research on this topic. Colin Price et al. [131,132] provided evidence for a link between natural ELF fields and ELF fields found in many organisms, including humans, proposing the notion of direct or indirect coupling of atmospheric electromagnetic systems to organisms. In addition, magnetic field oscillations caused by atmospheric events may have an impact on plant growth and development as well as on plant responses to other environmental factors. For example, when wheat and peas are exposed to an ELF magnetic field with SR frequency, their photosynthetic responses are affected to varying degrees depending on the plant species and exposure time [133].

## 5. Summary and Prospect

SR, as a natural ELF resonance excited by lightning activity in the Earth–ionosphere cavity, contains rich geophysical information. In terms of observation methods, it has expanded from traditional ground-based observation to satellite-based observation, and has been applied and studied in multiple fields such as lightning, earthquakes, and Earth's climate, and is constantly expanding. Therefore, the in-depth study of SR signals and their changing characteristics based on satellite data has become a development direction in future space geoscience.

It is worth noting that the range of sites for ground-based station observation will be gradually limited due to global rapid modernization, diverse industrial activities, and urban environments that cause a variety of ELF interferences, such as radiation from power lines, inductive signals from cell phone communication, leakage currents from grounding industrial objects, etc. We target our research in the field of satellite-based SR observations where these anthropogenic interference signals are expected to be eliminated as background noise in the spectrum of the satellite electromagnetic field.

On 2 February 2018, as the first satellite of the Chinese Geophysical Field Satellite Program and the first space-based platform of the China Earthquake Stereoscopic Observation System, the first China Seismo-Electromagnetic Satellite (CSES-01) was launched into orbit [134–136]. As shown in Figures 2–4, we identified and extracted the signal parameters

of the first two modes of SR using the electric field data of the CSES satellite, and plotted the corresponding SR background using all SR parameters in one revisit cycle (5 days).

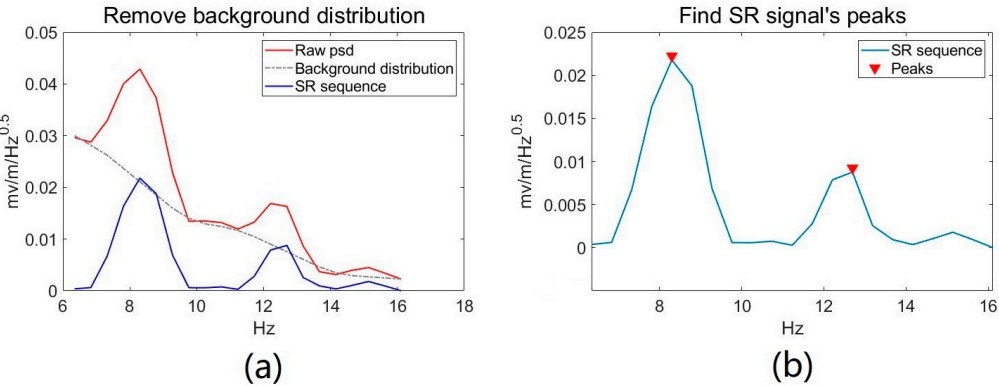

**Figure 2.** The extraction method of SR parameters: (**a**) represents the process of removing the background contribution of the original power spectrum of the electric field, and (**b**) shows the method of locating and extracting the SR parameters.

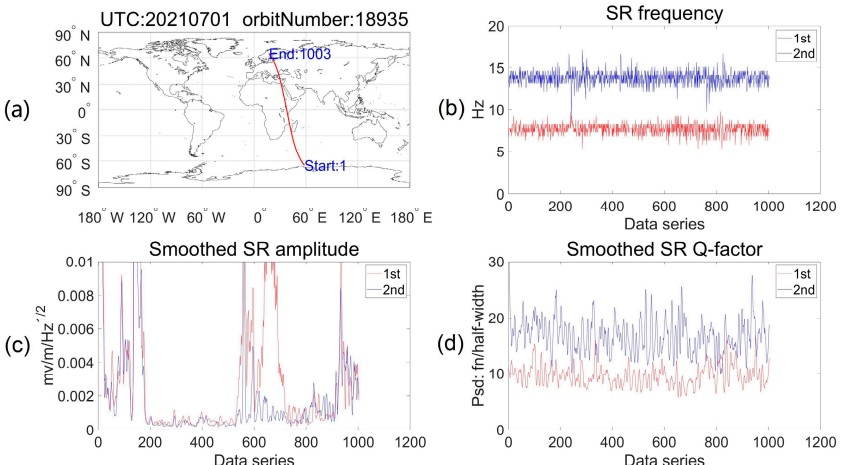

**Figure 3.** SR variation along the orbit at night: (**a**) shows the geographical position of the satellite orbit, (**b**–**d**) correspond to the SR frequency, amplitude, and quality factor extracted from this orbit, respectively. The red line represents SR's first mode and the blue line represents SR's second mode.

The methods shown in Figure 2 for removing the background contribution include separating and smoothing the background contribution and removing the smoothed background noise from the original power spectrum. We removed the background contribution of the electric field power spectrum and obtained the parameters of the first two modes of SR signals by identifying the peaks.

As shown in Figure 3, the amplitude trends of SR's two modes are similar, and the amplitude of SR's first mode is generally higher than the second mode. When passing through the central African region, the amplitude of SR signals extracted significantly increases (Figure 3a,c). Figure 3d shows that the Q factor also increases as the SR frequency increases. Based on the above methods and results, the SR background field is drawn as shown in Figure 4, which provides a solid foundation for further research on SR signal characteristics and their correlation with events such as earthquakes.

Currently, China's CSES-02 satellite is in development and will be launched in 2024. With more and more satellites collecting electric field and magnetic field data, SR research based on space-based observations will experience rapid expansion in the future.

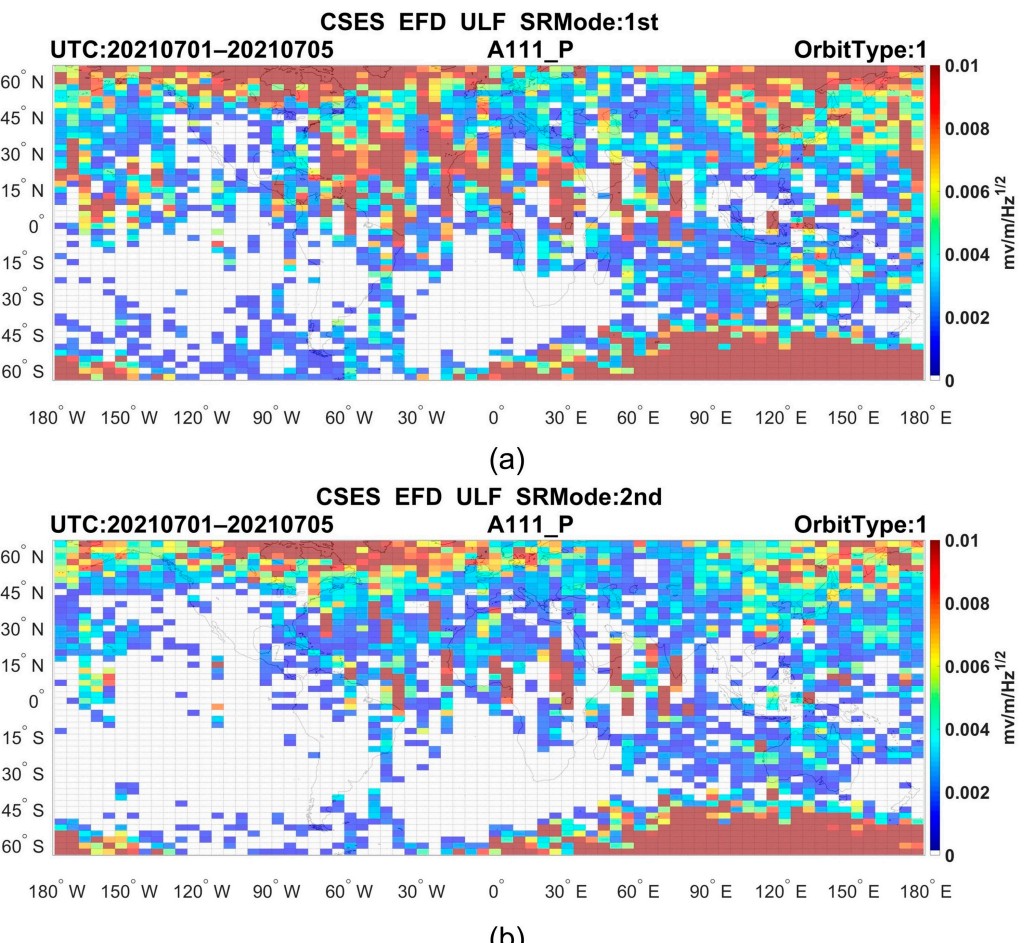

**Figure 4.** Global SR amplitude background: (**a**,**b**) represent the global background distribution of the amplitude of the first and second modes of SR, respectively.

**Author Contributions:** Conceptualization, J.L. and J.H.; methodology, J.L., J.H. and Z.L.; software, J.L., J.H. and Q.W.; validation, J.H., Z.L. and Z.Z. (Zhengyu Zhao); formal analysis, J.H.; investigation, J.L. and J.H.; resources, J.H., Z.Z. (Zhima Zeren) and X.S.; data curation, J.L.; writing—original draft preparation, J.L.; writing—review and editing, J.L., J.H. and Z.Z. (Zhengyu Zhao); visualization, J.L.; supervision, J.H.; project administration, J.H. and Z.Z. (Zhima Zeren); funding acquisition, Z.Z. (Zhima Zeren) and X.S. All authors have read and agreed to the published version of the manuscript.

**Funding:** This research was jointly funded by the National Natural Science Foundation of China (NSFC) Project, grant number 42104159; the Investigation of the Lithosphere Atmosphere Ionosphere Coupling (LAIC) Mechanism before the Natural Hazards, ISSI 23-583; and the Asia-Pacific Space Cooperation Organization earthquake special project phase 2, ISSI-BJ2019, Dragon 5 Cooperation Proposal, #58892, #59308.

**Data Availability Statement:** The data of CSES can be downloaded from the website (http://www.leos.ac.cn/). The data used by the article was accessed on 1 February 2023.

**Acknowledgments:** The authors acknowledge the International Space Science Institute (ISSI in Bern, Switzerland and ISSI-BJ in Beijing, China) for supporting International Team 23-583 lead by Dedalo Marchetti and Essam Ghamry, and wish to thank Ni Binbin and Ouyang Xinyan for their contributions during the SR study based on CSES data.

**Conflicts of Interest:** The authors declare no conflict of interest.

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
