# Peer review of "Recent Advances and Challenges in Schumann Resonance Observations and Research"

_remotesensing, doi:10.3390/rs15143557_

Round 1

Reviewer 1 Report

This paper summarized observations and applications of SR, and at the last, they used CSES data to show global distribution of SR. As a review paper, I found the last part is interesting and would like to know more.

 For the whole paper, there are two main concerns:

1) This paper has listed many articles, however, I did not get much new information that I couldn’t find elsewhere. Different/concise comments or review on literatures are needed.

2) Electromagnetic observations by Satellites to study on SR may be a new part which can make this paper different from previous review papers. As the SR signal is mainly concentrated in the Earth-ionosphere cavity, which is difficult to be seen in the topside ionosphere, and studies on SR by satellites are still limited. Section 5 gave first-order SR signals globally by CSES. To make this result more reliable, please introduce how do authors obtain SR signals, and show what it would be like along one orbit. Does CSES observe the typical SR signals?  

Author Response

Thank you for your comments and valuable suggestions. Our responses are attached, please see the attachment.

Reviewer 2 Report

I recommend the manuscript for the publication in its present form with a note: since it is a review paper, it could have been more elaborate. However, as such, all relevant materials are present in the manuscript.

Author Response

(The authors gave the same response as above.)

Reviewer 3 Report

Review of “Recent advances and challenges of Schumann resonances observatory and research”

By Jinlai Liu, Jian Ping Huang, Zhong Li, Zhengyu Zhao, Zhima Zeren, Xuhui Shen, Qiao Wang

This review article pulls together a selected collection of recent research results on Schumann resonances (SR) that even by itself is quite valuable in spreading the word about the extensive reach of this research topic into other areas of interest, including earthquake prediction of special interest to some of the authors.  The text is well organized but lacking in comprehensiveness of SR-related work (if this was intended). If one takes up the challenge of preparing a review article, then one ought to do it thoroughly. Was a thorough literature search undertaken (easy to do these days) in selecting topics for this work? The main shortcoming of the work is its inattention to theoretical models for SR and the two special dissipation heights pertaining to the electric field and the magnetic field, respectively.   Yes, the stated focus on “observatory” (observations) is noted, but without the expanded discussion on theory/models, other important areas of research are neglected (i.e., impacts of the Sun and from Space).  Last but not least, the authors are not English speakers and a great deal of editing is needed on this manuscript.

Summary: Consider for publication after major revision

Substantive issues:

(1)    Key missing reference

The very best paper on SR ever written (this reviewer’s opinion; the authors can comment when they have studied the paper) is not cited in this review:  Madden and Thompson (1965).  Yes, it is not “recent” but results therein underly so much else that is discussed by these authors.  Some examples:  first recognition of the two characteristic layers of dissipation, later investigated independently by Greifinger and Greifinger (1978).  This paper also gave first consideration to the nighttime leakage (a discovery incorrectly attributed to later authors) from the cavity which is essentially the source for what these authors refer to as SR in space.  (If coauthor Zhao Zhengyu found Schumann resonances at satellite altitude in 2000 (I am unable to read Chinese), then I hope they were found on the nightside of the Earth and not on the dayside.)  Also the effects of high altitude nuclear explosions on SR transient excitation were treated.  The most sophisticated modeling of the SR cavity ever undertaken, including tensor conductivity, is included there.

Also, the best current model for the uniform cavity (the so-called “knee model” of Mushtak and Williams (2002)) is not addressed, nor the theoretical finding that Q-factors for SR modes arse increasing with frequency, contradicting to some extent the statement in line 134 on page 3.  Detailed intercomparisons of models for the uniform cavity by Williams, Nickolaenko and Mushtak (2005) are neglected.  The textbook by Nickolaenko and Hayakawa (2002) does not do justice to model intercomparisons.  Without accurate model, the interpretation of SR observations may be flawed.

The day-night asymmetry of the SR cavity (addressed by M&T, 1965) brings important implications for the interpretation of SR observations, but unfortunately this topic is also not treated.

(2)    Referencing

The authors’ habit to separate the cited authors’ names from the reference number in brackets should be strongly discouraged.  These two indications should always come together.  Then there is no ambiguity about the actual citation intended.

(3)    Q-bursts

The only attention given to the transient aspect of SR appears in only 10 lines (246-256).  “Q-pulse trains” is also not standard terminology here.  Toshio Ogawa’s naming-of-phenomena and key contributions here are neglected.  Global geolocation of such events has been undertaken by Yamashita et al. (2011) based on four ELF stations.  These events should be of special interest to earthquake SR specialists because they can be used in the future for repeated probing of ionospheric properties in the epicentral regions, like the use a long-wavelength radar that is pulsed repeatedly (and irregularly in this case) by mesoscale convective systems in a fixed geolocation. (This gives the transients a big advantage over the background SR, which has been traditionally used in this context.)  The strong connection between mesoscale lightning and transient luminous events is also given meager attention, despite this connection giving a big boost to the SR research topic.  The work by Satori et al. (2016) on the impact of the variable Sun on the SR is also largely neglected.

(4)    Solar cycle effects

The SR cavity is strongly impacted over the 11-year solar cycle by variable solar x-radiation (e.g., Kulak et al., 2003; Satori et al., 2005) and by Energetic Electron Precipitation (EEP) (Bozoki et al., 2021).  These important processes seem to be summarized by a single sentence (lines 263-265).  But what then is the evidence that an “ionospheric density gradient” is involved?  The real physical basis for these effects needs to be fleshed out and clearly described.

(5)    ENSO impact

The important thermodynamic effects of El Nino are not adequately articulated. This is the most important source of interannual variability in the Earth’s atmosphere.  Recent work on Super El Nino events showing that the greatest lightning activity is apparent during the transition from the cold phase to warm phase is not addressed (Williams et al., 2021).

(6)    Other Chinese work

Recently, in reviewing a manuscript for the Journal of Geophysical Research, we were delighted to discover another group in China (The State Key Laboratory of Earthquake Dynamics) with dedicated efforts and abundant observational resources on SR, and including an ELF transmitter.  Given that the present authors are Chinese and given that this MDPI journal is largely Chinese, it seems reasonable that if publications exist from this group, that they ought to be discussed in the review.  Another consideration in this same context is the rapid development in China that is making it increasingly difficult to find low-noise observing stations for SR (Haiyan Yu, personal communication).

(7)    The Tonga volcanic eruption of January 15, 2022

The most exceptional volcanic eruption/lightning display on record occurred 16 months ago, with many publications involving dramatic SR effects, and it is not mentioned at all in this review.

(8)    Biomedical issues and SR

There has been an abundance of publications linking SR measurements with biological/medical issues.  C. Price in Tel Aviv has been an important contributor here.  This topic has been largely bypassed, despite the existence of a section in the review entitled “Application of SR in other fields”.

(9)    CSES Satellite data and the global map

An intriguing Figure 2 is embedded in the Summary and Prospect of this review. It looks to be very interesting, but altogether too little background information is included about how and when other data were acquired and how they were processed to make the global map.  I think it is also advisable that the work by Toledo-Redondo et al. (2012) based on DEMETER satellite observations also be cited and discussed.  This latter work is also discussed and re-interpreted in the paper by Bozoki et al. (2021) mentioned earlier.

See detailed review in the file attached.

Substantial editing is needed.   I have included my hand edits in a scanned copy of the manuscript that I will send separately to the Editor (Neil Zhang) since I do not see any option here to upload another file.

Author Response

(The authors gave the same response as above.)

Round 2

Reviewer 1 Report

1. For my first concern, the authors told me that they added more literatures, which is not my focus. Anyway, the review article is difficult, and it needs authors’ comprehensive understanding of the field.

2. For the second issue, the authors gave Figures 2 and 3 to show how they obtained SR signals. However, SR signals are phenomena in the frequency domain. Readers need to know whether SR signals can be seen in the spectrogram of electric field along the orbit. If they cannot be observed, it is not reasonable to obtain SR peak frequency, amplitude and Q-factor. In addition, we do not know the specific time of Figure2, which made it cannot be verified or repeated. Figure 3 show data on August 1, 2018, but Figure 4 show results in July, 2019. I suggest being consistent in the data analysis.

Author Response

Thank you again for your guidance! Please see the attachment.

Author Response

(The authors gave the same response as above.)
